# Clinical Application of a Customized 3D-Printed Bolus in Radiation Therapy for Distal Extremities

**DOI:** 10.3390/life13020362

**Published:** 2023-01-28

**Authors:** Suah Yu, So Hyun Ahn, Sang Hyoun Choi, Woo Sang Ahn, In-hye Jung

**Affiliations:** 1Department of Radiological Science, Kangwon National University, Samcheok 25949, Republic of Korea; 2Korea Institute of Radiological & Medical Sciences, Seoul 01812, Republic of Korea; 3Ewha Medical Research Institute, College of Medicine, Ewha Womans University, Seoul 07804, Republic of Korea; 4Department of Radiation Oncology, Gangneung Asan Hospital, University of Ulsan College of Medicine, Gangneung 25440, Republic of Korea

**Keywords:** customized 3D-printed bolus, distal extremities, dose build-up, reproducibility

## Abstract

In radiation therapy (RT) for skin cancer, tissue-equivalent substances called boluses are widely used to ensure the delivery of an adequate dose to the skin surface and to provide a radioprotective effect for normal tissue. The aim of this study was to develop a new type of three-dimensional (3D) bolus for RT involving body parts with irregular geometries and to evaluate its clinical feasibility. Two 3D-printed boluses were designed for two patients with squamous cell carcinoma (SCC) of their distal extremities based on computed tomography (CT) images and printed with polylactic acid (PLA). The clinical feasibility of the boluses was evaluated by measuring the in vivo skin dose at the tumor site with optically stimulated luminescence detectors (OSLDs) and comparing the results with the prescribed and calculated doses from the Eclipse treatment planning system (TPS). The average measured dose distribution for the two patients was 94.75% of the prescribed dose and 98.8% of the calculated dose. In addition, the average measured dose during repeated treatments was 189.5 ± 3.7 cGy, thus demonstrating the excellent reproducibility of the proposed approach. Overall, the customized 3D-printed boluses for the RT of distal extremities accurately delivered doses to skin tumors with improved reproducibility.

## 1. Introduction

Radiation therapy (RT) is a common form of cancer treatment primarily used when a tumor is large or widely distributed and difficult to remove using surgery [1]. In clinical practice, photon beams are used to customize the dose lesions that can appear with a variety of shapes, dimensions, and depths of invasion [2]. Photon beam energy is different depending on the treatment site. In the treatment of an irregular surface such as the fingers or toes, low energy in the kV range can lead to an inadequate tumor coverage and excess doses, thus high-energy photon beams in the MV range are generally employed to deliver a homogeneous dose distribution to the tumor [3]. Nevertheless, high-energy photon beams deliver a lower dose to the surface than to deeper tissue due to their high penetration and the consequent build-up effect, which has a skin-sparing effect that minimizes the dose delivered to superficial tumors [4,5]. As a result, in RT for shallow tumors, megavoltage photon beams have lower treatment effectiveness [6].

As a response to these challenges, the application of a bolus to the patient’s skin has been developed as a clinical solution to administer a sufficient dose to the skin surface by shifting the build-up region [7]. The most important characteristics of a bolus are its tissue-equivalence and malleability, which allows it to conform to the skin surface [8]. Currently, commercial boluses and various other tissue equivalents such as water, rice, and paraffin wax have been utilized for RT, but their insufficient malleability often results in unpredictable air gaps and secondary skin-sparing effects, thus reducing both the maximum and surface doses [9,10,11]. To overcome this drawback, the use of patient-specific boluses that are optimized for the treatment site has been proposed. In particular, three-dimensional (3D) printing technology has been used to create various structures based on digital models [12]. In general, two bolus fabrication methods based on 3D-printing technology have been employed in clinical practice [13]. The first method is directly printing a bolus using 3D-printing materials such as polylactic acid (PLA) and acrylonitrile butadiene styrene (ABS), which has the advantages of a shorter manufacturing time and a fixed treatment posture. The second method is to print the mold of the bolus and then fill it with another material, such as silicone rubber, to enhance the comfort of the patient.

Several studies have created individually customized 3D-printed boluses that conform to the complex anatomy of various body parts and have examined their clinical efficacy. For example, Baek et al. applied 3D boluses during the RT of patients with cancer of the oral cavity and the supraclavicular area. The authors showed that it was possible to reduce the dose delivered to the surrounding normal tissue by 14.4% while maintaining the target coverage [14]. In addition, Kim et al. treated patients with mycosis fungoides using a 3D bolus and accurately delivered doses that were within 0.3–2.1% of the planned treatment dose [3]. Kong et al. also fabricated a 3D bolus with hydrogel and simulated treatment with a head phantom; they found that the volume delivered 95% of the prescribed dose, an increase of 1.5% compared to a commercial bolus [15]. James et al. evaluated the use of a 3D-printed bolus for 16 patients with breast cancer and verified that it provided a 13–30% better fit to the chest wall than a sheet bolus [16].

In RT, it is important to accurately deliver the dose to the tumor only in order to reduce secondary cancer risks and improve survival [17]. Positioning reproducibility over the entire course of treatment is also an important characteristic for determining treatment outcomes [18]. Therefore, it is necessary not only to evaluate the target range after applying a 3D bolus but also to confirm its reproducibility during repeated treatment sessions. In this study, a new type of 3D bolus structure was developed for the treatment of patients with squamous cell carcinoma (SCC) in their distal extremities. To evaluate the positioning reproducibility and clinical feasibility of the proposed bolus in terms of its radioprotective effect for normal tissue, in vivo skin dose measurements with optically stimulated luminescence detectors (OSLDs) were taken and compared with the prescribed and calculated doses for a treatment planning system (TPS).

## 2. Materials and Methods

### 2.1. Patients and Treatments

In this study, a customized 3D-printed bolus was used on two patients with SCC in their distal extremities who were treated at Gangneung Asan Hospital. The patient characteristics are summarized in Table 1. The patients were treated with a 6 MeV high-energy photon beam. For Patient 1, the treatment was planned to be performed twice and a total of 240 and 265 monitor units (MU) were set to be delivered to the primary and secondary irradiation, respectively. For Patient 2, a total of 235 MU was set to be delivered. The planning of the 3D conformal radiation therapy (3D CRT) was performed using Eclipse version 16.1 (Varian, Palo Alto, Santa Clara, CA, USA) and treatment was performed using Varian Clinac iX (Varian, Palo Alto).

### 2.2. Manufacturing Workflow for the Customized 3D-Printed Boluses

The boluses were directly printed to control the build-up region and to ensure the immobilization of the treatment position. First, to create the 3D bolus structure, CT images of treatment area of both patients were acquired using a CT simulator (Discovery™ RT 16 channel, GE Healthcare, Chicago, IL, USA) with a scan thickness of 2.5 mm. A finger- and toe-shaped fixing frame with Styrofoam was created to establish the treatment area. For Patient 1, the CT simulation was run after inserting their finger into the Styrofoam frame. For Patient 2, a 4 cm thick solid water phantom was placed under a plastic bucket, and the patient’s foot was placed on a Styrofoam holder on top of the solid water phantom. Water was then poured into the plastic bucket and the CT simulation conducted.

As shown in Figure 1, the acquired CT images were sent to the TPS and extracted as Digital Imaging and Communications in Medicine (DICOM) files to design the 3D bolus structures.

The structure file for the 3D bolus was imported and converted to a stereolithographic (STL) file, which is commonly used for 3D printing, using 3D Slicer (version 4.10.0). The STL file was printed with Replicator™2 (MakerBot, Brooklyn, NY, USA). PLA was used as the filament material because PLA is a non-toxic material that has the advantage of maintaining its geometric integrity during printing with almost no shrinking upon cooling. The in-fill percentage was 98%, and a hexagonal print pattern was used to produce a highly rigid structure while minimizing the in-fill required.

To ensure a convenient set-up and to minimize the air gap between the skin surface and the bolus, the bolus for Patient 1 was divided into lower and upper parts, with the fixed parts for the hand and wrist indicated by the red arrows in Figure 2a. The bolus for Patient 2 was manufactured integrally as shown in Figure 2b. The fabricated 3D boluses were applied as shown in Figure 2, and CT simulation was performed for RT planning. A CT scan was taken in the same posture as during treatment and the treatment plan was performed based on this CT image. Patient 1 was positioned in a prone position on the treatment table during irradiation, while Patient 2 was in a sitting position. When irradiating the toe of Patient 2, a plastic bucket filled with water was used to make the skin surface flatter, but it was not used for the irradiation of the finger of Patient 1 in order to minimize postural inconvenience.

### 2.3. Dose Evaluation

To evaluate whether the prescribed dose was accurately delivered to the target site and how closely this delivered dose matched the calculated dose from the TPS, a total of four sites were selected for measurement and skin surface and in vivo dosimetry was conducted using the OSLDs. The measured dose was calculated as the mean of the maximum dose at each of the four points. This skin-surface dose was measured several times to analyze the variation in the target coverage and to verify the reproducibility of the treatment. In this study, the target coverage (%) was defined as the OSLD dose divided by the prescribed dose [15]. The site of attachment of the OSLD was determined within the tumor site in the first treatment, then marked and repeated. For Patient 1, the OSLD was placed on the upper bolus (Pos1) and the target region (Pos2) and measurements were taken 10 and 8 times, respectively. For Patient 2, the dose was measured 11 and 9 times with the OSLD under the base of the 5th toe (Pos1) and next to the left side of the 5th toe (Pos2), respectively. The OSLDs were calibrated at 200 MU with a single 6 MeV photon beam, 10 × 10 cm^2^, open field, a 100 cm surface to source distance (SSD), and on the central axis of the beam. After that, they were compared with ion chamber (FC65-P) measurements, with individual correction factors applied.

## 3. Results

### 3.1. Dose Evaluation

The measured doses for Patients 1 and 2 are presented in Table 2 and Table 3, respectively. For Patient 1, the average and deviation measured dose for the upper 3D bolus and the target region was 193.2 ± 4.1 cGy and 187.6 ± 2.1 cGy, respectively, which represented excellent agreement and was confirmed with approximately 99.3% and 98.0%, respectively, of the calculated dose from the TPS. For Patient 2, the average and deviation measured dose for the base of the 5th toe and the left side of the 5th toe was 189.2 ± 3.7 cGy and 187.7 ± 4.9 cGy, respectively, which represented excellent agreement and was confirmed with approximately 99.1% and 98.6%, respectively, of the calculated TPS dose.

### 3.2. Target Coverage

The target coverage at Pos1 and Pos2 for Patients 1 and 2 was also calculated (Figure 3). For Patient 1, the average target coverage was 99.1% and 100.8% at Pos1 and Pos2, respectively, with a deviation of 2.3% and 1.9%, respectively. For Patient 2, the average target coverage was 99.6% at Pos1 and 98.9% at Pos2, with a deviation of 1.8% and 2.1%, respectively. On average, a dose close to the prescribed dose was accurately delivered to the tumor area within an error of 3%, and the standard deviation was also within 3%, confirming that the reproducibility of the proposed treatment method was excellent during repeated treatment in both patients.

## 4. Discussion

In RT for skin cancer, modern treatment methods such as intensity-modulated RT, volumetric modulated arc therapy (VMAT), or tomotherapy can deliver doses near the skin without exposing normal tissue to the radiation. Dias et al. confirmed that the effect of the use of a female anthropomorphic Rando phantom on the skin dose was considerable, with less radiation delivered to organs at risk than 3D CRT. In addition, in the treatment of a patient with breast carcinoma using VMAT, the use of a bolus optimized the positioning reproducibility [19]. However, for distal extremities such as fingers and toes, the dose distribution is more strongly affected by inter-fraction motion due to their small, irregular, and complex shape. Therefore, a bolus is necessary to induce an appropriate dose build-up effect [20].

When using megavoltage photon beams to treat skin cancer, a bolus should be employed to reduce the skin-sparing effect and generate a dose build-up on the skin. The curved surface of the treatment area makes it difficult to completely match the skin and bolus, creating an air gap. This air gap creates a secondary skin-sparing effect and reduces both the surface and maximum doses. In addition, patient movement shifts the position of the air gap, causes random hot spots, and makes it difficult to deliver an accurate dose. Therefore, the bolus should fit the surface being treated as much as possible and movement should be minimized by immobilizing the patient [21].

Customized 3D-printed boluses have been developed to produce a more accurate dose distribution than conventional boluses and to improve the reproducibility of the treatment. Many studies have consequently demonstrated a reduction in the air gap and an improvement in the accuracy of the dose build-up, while also reducing the time and money required. However, there have been few reports of these boluses being employed in clinical practice, and past research has rarely considered the characteristics of repetition of RT sessions [22,23].

Therefore, in the present study, we repeatedly measured the actual dose applied to the target areas of two patients to evaluate the accuracy of the dose build-up and its reproducibility. For Patient 1, the skin dose was higher than the prescribed dose, and the accuracy compared with the TPS was 99.3%. For Patient 2, the skin dose was higher than the prescribed dose except for one measurement and was 98.9% accurate. Therefore, the proposed customized 3D-printed bolus was found to accurately deliver the prescribed dose to the skin and to overcome the skin-sparing effect.

The hand and wrist of Patient 1 were placed in the bolus, while the skin surface of Patient 2 was made as flat as possible by placing their foot in water, which minimized the patient’s movement during treatment. It was found that the deviation in the dose for Patients 1 and 2 was 5.6 cGy and 4.2 cGy, respectively, indicating excellent reproducibility. However, the usefulness of the customized 3D-printed bolus developed in this study was applied to only two patients, so there is a need for further research to provide sufficient data for generalization.

## 5. Conclusions

In the present study, a new type of customized 3D-printed bolus was developed to overcome the limitations of conventional boluses for RT for skin cancer in the distal extremities, and its usefulness was evaluated by applying it to clinical cases. A dose with a difference of 3% from the prescribed dose was delivered to the tumor site and there was a variation of 3% in the dose over the entire treatment process. Therefore, the use of a customized 3D-printed bolus can improve reproducibility by minimizing the movement of the treatment site and increase treatment efficiency by delivering a maximum dose to the tumor.

## Figures and Tables

**Figure 1 life-13-00362-f001:**
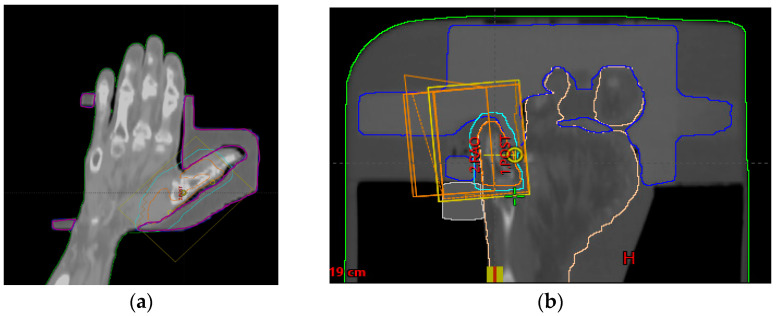
CT simulation frontal view of patient 1 and 2: (**a**) Treatment area for Patient 1. The purple line is the form of the 3D bolus; (**b**) Treatment area for Patient 2. The blue line is the form of the 3D bolus.

**Figure 2 life-13-00362-f002:**
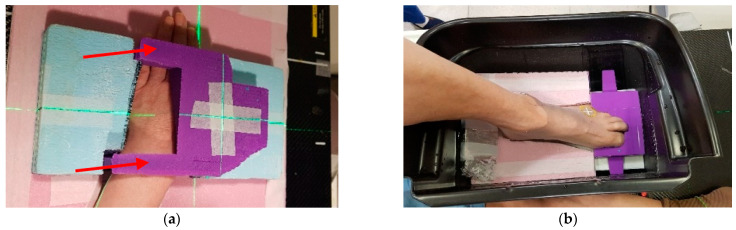
Fabricated bolus and patient set up for radiation therapy: (**a**) 3D bolus for patient 1. The upper arrow is the fixed part of the hand, and the lower arrow is the fixed part of the wrist; (**b**) 3D bolus for patient 2. The plastic bucket was filled with water to make the surface flatter.

**Figure 3 life-13-00362-f003:**
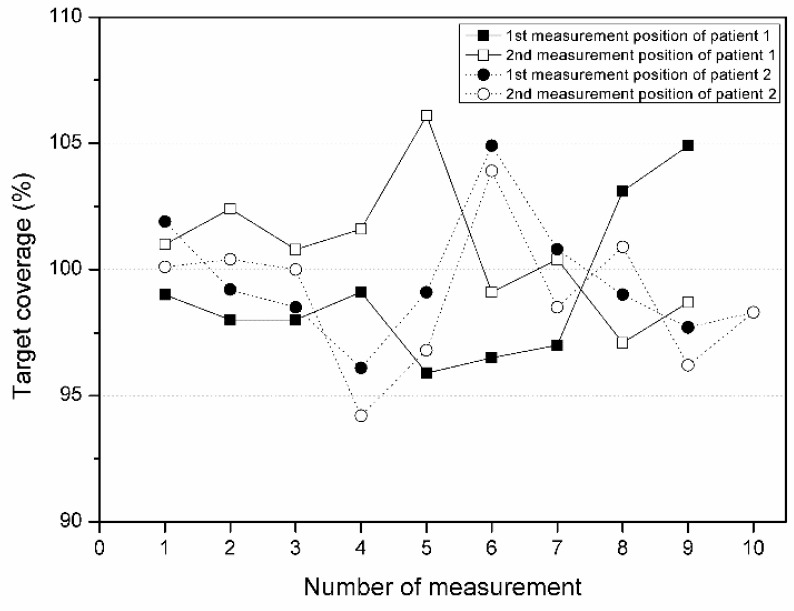
Target coverage of patients 1 and 2.

**Table 1 life-13-00362-t001:** Patient characteristics.

	Patient 1	Patient 2
Gender	Female	Male
Age	66 y	68 y
Prescribed dose	60.4 Gy/33 fx (50.4 Gy/28 fx, 10 Gy/5 fx)	54 Gy/30 fx
MU	240, 265	235
Treatment site	Left 1st metacarpal phalangeal joint	Right 5th toe
Beam energy	6 MeV	6 MeV

**Table 2 life-13-00362-t002:** In vivo dosimetry with the OSLD measurement results of patient 1.

	OSLD Measurement Mean Dose (cGy)	Difference to Prescription	Difference to TPS
Upper 3D bolus (Pos1)	193.2 ± 4.1	7.4%	−0.7%
Target region (Pos2)	187.6 ± 2.1	4.2%	−2.0%
Mean dose (cGy)	190.4 ± 3.1	5.8%	−1.4%

**Table 3 life-13-00362-t003:** In vivo dosimetry with the OSLD measurement results of patient 2.

	OSLD Measurement Mean Dose (cGy)	Difference to Prescription	Difference to TPS
Base of the 5th toe (Pos1)	189.2 ± 3.7	5.1%	−0.9%
Left side of the 5th toe (Pos2)	187.7 ± 4.9	4.3%	−1.4%
Mean dose (cGy)	188.5 ± 4.3	4.7%	−1.2%

## Data Availability

Not applicable.

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
