# Peer review of "Clinical Application of a Customized 3D-Printed Bolus in Radiation Therapy for Distal Extremities"

_life, 2023, doi:10.3390/life13020362_

Round 1

Reviewer 1 Report

Dear Authors,

topic of the manuscript is interesting and relevant; however, there are some major remarks, which should be defined, corrected, commented or/and explained (see attached file).

Author Response

Dear Reviewers

Thank you for giving us the opportunity to improve the quality of the manuscript through the quick and detailed comments of the reviewers. We carefully responded to the reviewers' comments and reflected them in the text. And the detailed answer to each comment is like the attached file.

Sincerely,

Suah Yu

Reviewer 2 Report

The authors of the article developed a new type of customized 3D-printed bolus to improve the limitations of the conventional bolus for skin cancer radiotherapy. The present work evaluated its usefulness by applying it to 2 clinical cases.

The newly developed bolus completely immobilized the patient, minimizing movement of the treatment site and assured good contact with the skin surface. The customized 3D-printed bolus improved the treatment efficiency by increasing reproducibility during repeated treatments and accurately delivering the dose to reduce the exposure of surrounding tissues. 

The customized 3D-printed bolus generated a more accurate dose distribution than the conventional bolus and improved the outcome also based on financial and time aspects.

It is a valuable case presentation, but the main limitation of this study is that the usefulness of the customized 3D-printed bolus was applied to only two patients, so there is a need for further studies that can provide sufficient data for generalization.

I also consider it important to add more data in the introduction section, in order to establish the state of the art in bolus RT at the moment, for readers that are not involved in the clinical aspects of this area of research.

Author Response

Dear Reviewers,

Thank you for giving us the opportunity to improve the quality of the manuscript through the quick and detailed comments of the reviewers. We carefully responded to the reviewers' comments and reflected them in the text. And the detailed answer to each comment is like the attached file.

Sincerely,

Suah Yu

Round 2

Reviewer 1 Report

Thank you for the corrections and explanations, but still here it is some minor remarks, which should be corrected, commented or/and explained (attached file).

Author Response

(The authors gave the same response as above.)
